

# Correlation between antiferromagnetic and charge-density-wave order in UPt$_2$Si$_2$ studied by resonant X-Ray scattering

Fusako Kon[1*], Chihiro Tabata[2], Kodai Miura[1], Ruo Hibino[1], Hiroyuki Hidaka[1], Tatsuya Yanagisawa[1], Hironori Nakao[3] and Hiroshi Amitsuka[1]

**1** Graduate School of Science, Hokkaido University, Sapporo 060-0810, Japan
**2** Materials Sciences Research Center, Japan Atomic Energy Agency, Tokai 319-1195, Japan
**3** Photon Factory, Institute of Materials Structure Science, KEK, Tsukuba 305-0801, Japan

⋆ 23kon@phys.sci.hokudai.ac.jp

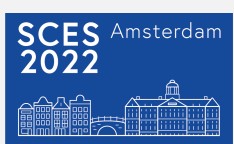

*International Conference on Strongly Correlated Electron Systems
(SCES 2022)
Amsterdam, 24-29 July 2022*

## Abstract

Relationship between antiferromagnetic order with Néel temperature of 35 K and charge-density-wave (CDW) order below 320 K of UPt$_2$Si$_2$ has been investigated by resonant X-ray scattering measurements. We have found that a resonant scattering signal develops below the Néel temperature at the $M_4$ absorption edge of uranium with a propagation vector of the CDW order $q_{\text{CDW}} \sim (0.42, 0, 0)$. The experimental results reveal that the magnetic structure of UPt$_2$Si$_2$ is modulated by the CDW ordering.

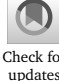
## 1 Introduction

The intermetallic uranium compound UPt$_2$Si$_2$ crystallizes in the tetragonal CaBe$_2$Ge$_2$-type structure (No. 129, $P4/nmm$, $D_{4h}^7$) [1] (Fig. 1), which has spatial inversion symmetry although none of the ions are located on the inversion center. This crystal structure can also be viewed as a layered structure, i.e., layers of uranium ions are stacked along the $c$-axis, sandwiched between two different layers: Pt(1)-Si(1)-Pt(1) (hereafter called layer-1) and Si(2)-Pt(2)-Si(2) (layer-2). The system is known to exhibit a collinear antiferromagnetic (AFM) order at $T_N \sim 35$ K with the propagation vector $\boldsymbol{Q} = 0$, where the magnetic moments ($\sim 1.7$ $\mu_B$/U) on the inversion pair of U ions in the unit cell are antiferromagnetically aligned along the $c$-axis (Fig. 2) [2,3]. The inversion center, located at the middle point of a U pair in paramagnetic phase, is lost under this staggered order of magnetic moments. Thus, one of the interesting features of this system is the loss of global spatial inversion symmetry due to the AFM ordering.

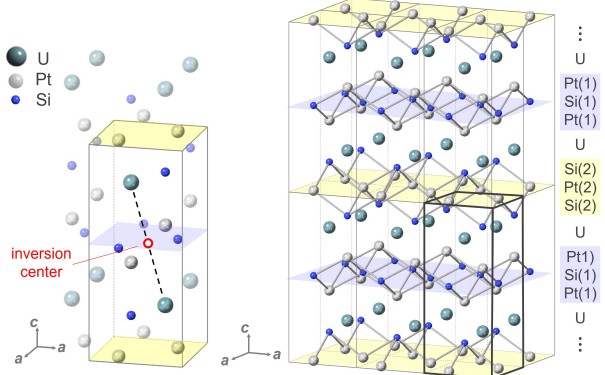

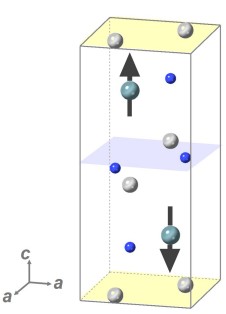

Figure 1: Crystal structure of UPt$_2$Si$_2$ and primitive unit cell (left panel). Multiple unit cells are depicted to highlight the layered structure (right panel). The blue and yellow colored regions represent layer-1 and layer-2, respectively.

Figure 2: Magnetic structure of UPt$_2$Si$_2$ with $\boldsymbol{Q} = 0$ reported in previous studies [2, 3].

Another notable feature of this system, which is the focus of this study, is a charge density wave (CDW) state. It is commonly found in several Pt-based systems with CaBe$_2$Ge$_2$-type structure. According to the band calculations for LaPt$_2$Si$_2$ [4, 5] and SrPt$_2$As$_2$ [5], they have cylindrical Fermi surfaces from the Pt(2)-$5d$ band and these quasi-nesting features may be responsible for the CDW transitions. UPt$_2$Si$_2$ also exhibits a CDW state below $T_{\text{CDW}} \sim 320$ K, which has recently been found by Lee *et al.* in neutron scattering and synchrotron non-resonant X-ray scattering experiments [6]. They suggest that the CDW wave vector $\boldsymbol{q}_{\text{CDW}} = (\tau,$ 0, 0) varies with decreasing temperature from a commensurate value of $\tau = 0.40$ just below $T_{\text{CDW}}$ and locks in to an incommensurate value of $\tau \sim 0.42$ below 180 K. They also found that the CDW phase coexists with the AFM phase below $T_{\text{N}}$. Their density functional theory calculations taking into account the observed diffraction patterns suggest that the CDW order is caused by the $5d$ electrons of the Pt ions in the layer-2 as in LaPt$_2$Si$_2$ and SrPt$_2$As$_2$. It is therefore of interest whether there is a correlation between the CDW order by Pt $5d$ electrons and the AFM order by U $5f$ electrons, which has not yet been investigated in detail. In this study, we investigate the $5f$ electronic states of U using resonant X-ray scattering (RXS), and show that the AFM-ordered structure is modulated with the period of the CDW order.

## 2 Experimental procedure

We have performed the RXS experiments at BL-11B of KEK Photon Factory using an in-vacuum two-circle diffractometer [7]. A single crystalline sample of UPt$_2$Si$_2$ was grown by the Czochralski method using a tetra-arc furnace at Hokkaido University. The sample was shaped into rectangular parallelepiped of $\sim 1 \times 3 \times 3$ mm$^3$ with polished wide plane ((100) plane). All measurements in this study are performed in the scattering plane of (001) in temperature range of 6-300 K by using $^4$He-flow cryostat. The incident beams were monochromatized with Si (111) double crystals. The X-ray energy was tuned around 3.72 keV (U $M_4$-edge: $3d_{3/2} \rightarrow 5f$ in the $E1$ transition) and polarized in the scattering plane ($\pi$-polarization). Polarization analysis of the scattered X-ray was performed using an Al (111) crystal.

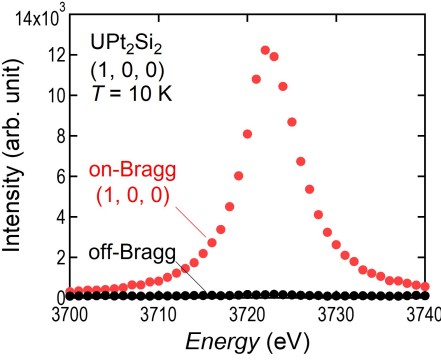

Figure 3: Energy dependence of (1, 0, 0) reflection intensity at 10 K (in AFM phase). Black filled circles indicate the data in off-Bragg condition (background).

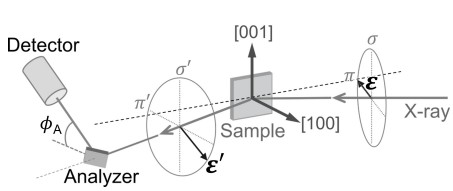

Figure 4: Scattering geometry of the polarization analysis. The polarization of the scattered X-ray is analyzed by rotating the polarizer scattering plane by angle $\phi_A$.

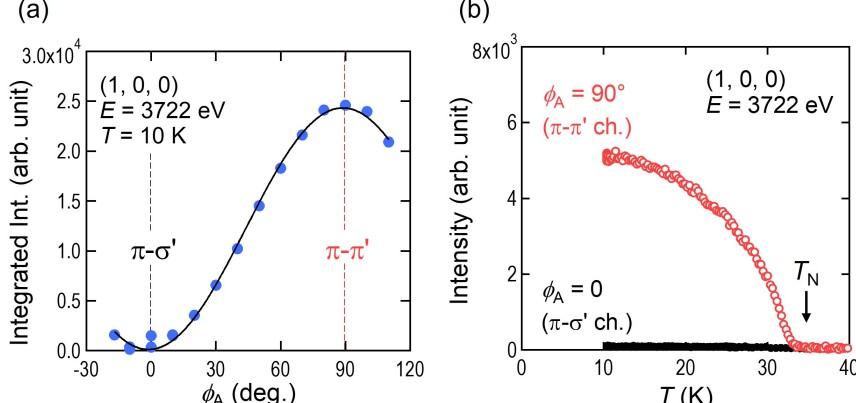

Figure 5: (a) Polarization dependence of the (1, 0, 0) reflection intensity with the scattering plane perpendicular to the *c*-axis. The solid curve represents the fitting results for the perfect $\pi$-$\pi$' scattering. (b) Temperature dependence of the (1, 0, 0) reflection intensity in $\pi$-$\pi$' channel (red open circles) and $\pi$-$\sigma$' channel (black filled circles).

## 3 Experimental results and discussions

Firstly, we checked the magnetic reflections with $Q = 0$ reported in the previous neutron scattering studies [2, 3]. In the AFM phase, we observed a series of reflections $(h, k, 0)$ $(h + k = $ odd), which do not satisfy the reflection conditions on the $P4/nmm$ space group. The energy dependence of (1, 0, 0) reflection intensity is shown in Fig. 3. A clear resonant spectrum can be seen at 3.722 keV, the energy of U $M_4$ absorption edge, confirming the order of the U $5f$ electrons that breaks $n$-glide symmetry of the crystal structure of the $P4/nmm$ space group. We investigated the polarization dependence of the (1, 0, 0) reflection intensity in the (001) scattering plane for the $\pi$-polarized incident beam. Figure 5 (a) shows the observed dependence of the (1, 0, 0) reflection intensity on the polarizer scattering angle $\phi_A$ (the scattering geometry is illustrated in Fig. 4). When the incident X-rays are perfectly $\pi$-polarized, the scattering intensities in $\pi$-$\pi$' and $\pi$-$\sigma$' scattering processes are calculated as $I_{\pi \to \pi'} \propto (1 - \cos 2\phi_A)$ and $I_{\pi \to \sigma'} \propto (1 + \cos 2\phi_A)$ [8]. The experimental data are in good agreement with the change assuming that the reflection occurs only in the $\pi$-$\pi'$ process. The intensity of the $\pi$-polarized scattered X-rays decreases with increasing temperature and disappears at $T_N$ (Fig. 5 (b)). Since

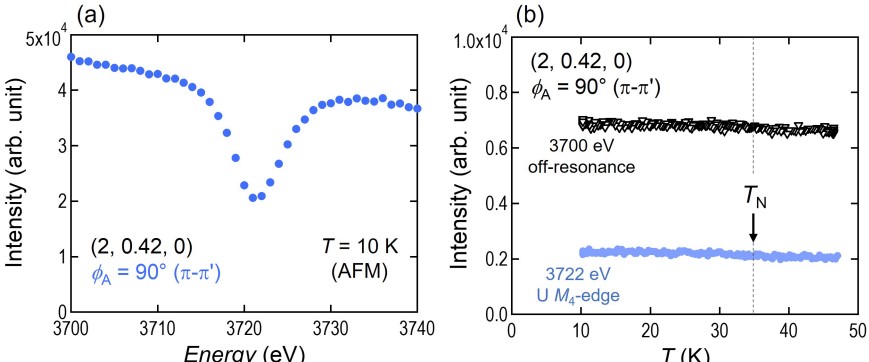

Figure 6: (a) Energy dependence of (2, 0.42, 0) reflection intensity in $\pi$-$\pi$' channel. (b) Temperature dependences of (2, 0.42, 0) reflection intensity in $\pi$-$\pi$' channel at the resonant energy (blue filled circles) and off-resonance (black open triangles).

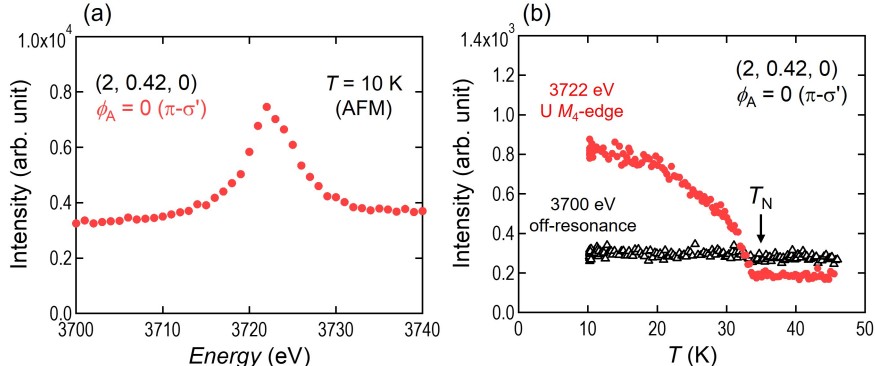

Figure 7: (a) Energy dependence of (2, 0.42, 0) reflection intensity in $\pi$-$\sigma'$ channel. (b) Temperature dependenes of (2, 0.42, 0) reflection intensity in $\pi$-$\sigma'$ channel at the resonant energy (red filled circles) and off-resonance (black open triangles).

the RXS amplitude due to a ordered magnetic moment $\mu_{ord}$ is proportional to $(\varepsilon' \times \varepsilon) \cdot \mu_{ord}$, where $\varepsilon$ ($\varepsilon'$) denotes the polarization vector of the incident (scattered) X-ray [9], a magnetic signal in the $\pi$-$\pi'$ channel is caused by the component of $\mu_{ord}$ perpendicular to the scattering plane, while one in the $\pi$-$\sigma'$ comes from the projection of $\mu_{ord}$ onto the scattering plane. In the present experimental configuration with the scattering plane (001), the fact that the signal is observed in the $\pi$-$\pi'$ channel and not in the $\pi$-$\sigma'$ channel means that $\mu_{ord}$ is parallel to the $c$-axis. Hence, we confirmed that the $\boldsymbol{Q} = 0$ component of spacial modulation in magnetic density, which is consistent with the previous reports [2, 3], was observed as the intensive resonance signal at the U $M_4$ absorption edge on UPt$_2$Si$_2$.

In addition to the resonant scatterings corresponding to the AFM order, we observed a set of resonant (3722 eV) and non-resonant (3700 eV) scatterings at the positions of Bragg conditions for the reported CDW propagation vector $\boldsymbol{q}_{CDW} \sim$ (0.42, 0, 0). Figures 6 and 7 show a typical example of one such scattering signal, plotting the energy and temperature variations of the scattering intensity for the scattering wave vector (2, 0.42, 0) in the energy range near the U $M_4$-edge. The energy spectrum in the $\pi$-$\pi'$ channel (Fig. 6 (a)) shows a large scattering intensity over the entire measured energy range, with an absorption of about 50% at $\sim$3.72 keV. This energy dependence is typical of non-resonant signals derived from crystal lattices. The scattering intensities at resonant and non-resonant energy do not change at $T_N$, as shown in Fig. 6 (b). The scattering observed in the $\pi$-$\pi'$ channel can therefore be attributed to Thomson scattering due to the crystal lattice modulated by the CDW order with the atomic displacements confirmed by Lee $et$ $al.$ [6].

By contrast, the energy spectrum in the $\pi$-$\sigma'$ channel, which is in principle purely caused by an order of magnetic dipoles (or higher-rank multipoles), exhibits a resonance-like enhancement at the absorption edge energy as shown in Fig. 7 (a). Here we note that the finite signal intensity was also observed at off-resonance energies such as 3700 eV, due to a small contamination of the $\sigma$-polarized incident X-rays causing a $\sigma$-$\sigma'$ scattering (Thomson scattering) from the crystal lattice. The signal intensity at the $M_4$-edge energy increases below $T_N$, while the off-resonance signal shows no temperature variation (Fig. 7(b)). This temperature dependence, together with the peak in the energy spectrum, indicates that the resonance scattering is caused by the spatially modulated magnetic order of U with the same period as the CDW order. Since only the in-plane component of $\mu_{ord}$ contributes to the $\pi$-$\sigma'$ scattering, the magnetic modulation observed in $q_{CDW}$ is considered to be caused by a slight cant of $\mu_{ord}$ from the $c$-axis in the AFM state. Thus, the AFM state of this system is not a simple $Q = 0$ magnetic structure as previously reported, but a more complicated structure in which the magnetic dipole components in the (001) plane are modulated with the $q_{CDW}$.

Several materials exhibiting both CDW and magnetic orderings are known in $f$-electron systems, such as $Er_5Ir_4Si_{10}$ [10], $RNiC_2$ ($R = $ Tm, Er, Ho, Dy, Tb, Gd, and Sm) [11–13], and $TbTe_3$ [14]. A common feature of these systems is that the CDW and magnetic order are governed by conduction electrons and localized $4f$ moments, respectively. The interplays between these orders, however, have not yet been studied in detail. In $TbTe_3$, Chillal *et al.* revealed that the CDW order in Te-layer may influence the AFM order of Tb $4f$ moments via the modulation of crystalline electric fields (CEF) states of $Tb^{3+}$ ions. In the case of $UPt_2Si_2$, it is sugguested that the CDW order is formed in layer-2 by Pt $5d$ electrons and accompanied by the atomic displacements at Pt(2) sites [6]. The present experimental results indicate that U $5f$ electrons are influenced by the CDW order, implying that the local symmetry felt by $5f$ electrons is spatially modulated. The interplay between both the orders must be mediated by CEF in a broad sense, including electrostatic potentials and the hybridization effects between $5f$ and $5d$ electrons as discussed in $TbTe_3$. For further discussion, it is needed to measure and analyze higher harmonic components and resonant scattering of Pt $5d$ electrons, which is currently in progress.

## 4 Conclusion

We revealed the interplay between magnetic order and CDW in $UPt_2Si_2$ by resonant X-ray scattering. Specifically, the magnetic structure contains not only collinear $Q = 0$ component with the magnetic moments along the $c$-axis but also an in-plane component with a spatial modulation described by $q_{CDW} \sim$ (0.42, 0, 0). One possible origin of the magnetic moment tilting from the $c$-axis would be the periodic modulation of the CEF effect. Although details are expected in future studies, we would like to point out that $UPt_2Si_2$ may be a suitable system for studying the effect of ligands on $5f$ electrons by regarding CDW order as a perturbative internal field.

## Acknowledgements

This work is partly supported by JSPS KAKENHI Grant Numbers JP15H05882, JP15H05883 (J-Physics) and JP21KK0046. The authors would also like to thank Prof. H. Kusunose and Prof. H. Harima for their helpful discussions.

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
