# Peer review of "Correlation between Antiferromagnetic and Charge-Density-Wave Order in UPt$_{2}$Si$_{2}$ Studied by Resonant X-Ray Scattering"

_SciPost Physics Proceedings, doi:SciPost Phys. Proc. 11, 011 (2023)_

## Round 1 · Referee Report · Anonymous (Referee 1) · 2023-1-4

Strengths

  • very well written.
  • Clear figures.
  • Demonstrates coupling between Collin ear AFM and CDW order.

Weaknesses

  • Not highly innovative

Report

The paper clearly meets the conditions for SciPost proceedings.

Requested changes

This is a well written and clear paper. I only have a few small remarks. 1: It would be useful if the authors describe the nature of the CDW order in the introduction. 2: The explanation of the determination of the magnetic order (first paragraph, page 4) is somewhat confusing. For both pi-pi and pi-sigma the authors seem to suggest that mu_ord projects onto the [001] plane.

  • validity: good
  • significance: ok
  • originality: ok
  • clarity: high
  • formatting: good
  • grammar: excellent

Author:  Fusako Kon  on 2023-01-07  [id 3217]

(in reply to Report 1 on 2023-01-04)
Category:
correction

Dear Referee:

We are very grateful for your kind suggestions and constructive advice for our manuscript scipost_202208_00025v1 by F. Kon et al. We would like to revise our manuscript according to your comments. Please confirm a summary of the changes:

Requested changes #1 It would be useful if the authors describe the nature of the CDW order in the introduction.

Our reply #1: We really appreciate your productive comment. Following your advice, we would like to add some sentences referring to the possible origin and the property of this CDW order in the introduction as follows.

1) p.2 “1. Introduction”, 2nd paragraph, l. 1- <before revision> Another notable feature of this system, which is the focus of this study, is a charge density wave (CDW) state below T_CDW ∼ 320 K, which has recently been demonstrated by Lee et al. by neutron scattering and synchrotron non-resonant X-ray scattering experiments [4]. This CDW phase has been found to coexist with the AFM phase below T_N. Their density functional theory calculations taking into account the observed diffraction patterns suggest that the CDW order is caused by the 5d electrons of the Pt ions in the layer-2.

<after revision> Another notable feature of this system, which is the focus of this study, is a charge density wave (CDW) state. It is commonly found in several Pt-based systems with CaBe2Ge2-type structure. According to the band calculations for LaPt2Si2 [4, 5] and SrPt2As2 [5], they have cylindrical Fermi surfaces from the Pt(2)-5d band and these quasi-nesting features may be responsible for the CDW transitions. UPt2Si2 also exhibits a CDW state below T_CDW ∼ 320 K, which has recently been found by Lee et al. in neutron scattering and synchrotron nonresonant X-ray scattering experiments [6]. They suggest that the CDW wave vector q_CDW =(τ, 0, 0) varies with decreasing temperature from a commensurate value of τ = 0.40 just below TCDW and locks in to an incommensurate value of τ ∼ 0.42 below 180 K. They also found that the CDW phase coexists with the AFM phase below T_N. Their density functional theory calculations taking into account the observed diffraction patterns suggest that the CDW order is caused by the 5d electrons of the Pt ions in the layer-2 as in LaPt2Si2 and SrPt2As2.

Requested changes #2 The explanation of the determination of the magnetic order (first paragraph, page 4) is >somewhat confusing. For both pi-pi and pi-sigma the authors seem to suggest that mu_ord >projects onto the [001] plane. Our reply #2: We fully agree with you. To clarify that the signal of pi-pi channel reflects the components of magnetic moments along the c-axis (not the components in the c-plane), we would like to revise our description as follows:

2) p. 4 “3. Experimental results and discussions”, 1st paragraph, l. 14- <before revision> Since the RXS amplitude due to a ordered magnetic moment µ_ord is proportional to (ϵ′ × ϵ) · µ_ord, where ϵ (ϵ′) denotes the polarization vector of the incident (scattered) X-ray [7], the observed signal in the π-π′ channel is caused by the component of µord, perpendicular to the scattering plane (here (001) plane). In contrast, the signal intensity in the π-σ′ channel, which is caused by the projection onto the (001) plane of µ_ord, was below the detection limit throughout the measurement temperature range.

<after revision> Since the RXS amplitude due to a ordered magnetic moment µ_ord is proportional to (ϵ′ × ϵ) · µ_ord, where ϵ (ϵ′) denotes the polarization vector of the incident (scattered) X-ray [9], a magnetic signal in the π-π′ channel is caused by the component of μ_ord perpendicular to the scattering plane, while one in the π-σ′ comes from the projection of μ_ord onto the scattering plane. In the present experimental configuration with the scattering plane in (001), the fact that the signal is observed in the π-π′ channel and not in the π-σ′ channel means that μ_ord is parallel to the c-axis.

We trust that the Referees will find the revisions of the manuscript improved satisfactorily and worth publication in SciPost Physics Proceedings.

Sincerely yours, Fusako Kon, Hiroshi Amitsuka, and all the co-authors

---

## Editorial Decision

published